# A comparative analysis of the dendritic cell response upon exposure to different rabies virus strains

**Keshia Kroh, Merel R. te Marvelde, Lars W. van Greuningen, Brigitta M. Laksono, Marion P. G. Koopmans, Thijs Kuiken, Corine H. GeurtsvanKessel[◐], Carmen W. E. Embregts[◐]***[◐]

Department of Viroscience, Erasmus Medical Center, Rotterdam, The Netherlands

◐ These authors contributed equally to this work.
* c.embregts@erasmusmc.nl

## Abstract

Rabies is a viral zoonotic disease that causes over 60,000 human deaths annually worldwide. Natural infections lack a virus-specific immune response, leading to a near 100% fatality rate unless immediately treated. Rabies virus (RABV) is typically transmitted through bites from rabid dogs or other carnivores to humans and may initially interact with innate immune cells such as dendritic cells at the site of infection. This study investigates the *in vitro* response of human monocyte-derived dendritic cells (moDCs) exposed to two pathogenic RABV strains—silver-haired bat rabies virus (SHBRV) and dog-related rabies virus (dogRV)—and an attenuated vaccine strain (SAD P5). MoDCs were susceptible only to high doses of SHBRV and SAD P5, resulting in a more mature and migratory phenotype within the infected moDC populations. No infection was observed in moDCs exposed to dogRV. In co-culture with T cells, the presence of RABV-exposed moDCs, regardless of the strain, did not enhance T cell activation. Additionally, RABV exposure did not hinder LPS-induced moDC maturation; instead, high doses of SHBRV and SAD P5 even boosted activation levels. Overall, the findings suggest varied capabilities of RABV strains to infect and activate moDCs *in vitro*. However, exposure to any RABV strain did not provoke a clear antiviral state or suppression of moDC responsiveness. This lack of activation may contribute to the absence of an effective adaptive immune response in natural RABV infections.

## Author Summary

Rabies virus infections in humans are almost always fatal if left untreated due to a lack of a protective immune response. Dendritic cells play a crucial role in initiating an antiviral immune response, so our study investigated their response to exposure to three different rabies virus strains. We discovered that two of the three strains could infect a small proportion of dendritic cells. The third strain did not activate the cells, while exposure to the other two strains led to only minor signs of activation. This response was insufficient to trigger a significant immune reaction. Despite exposure to the virus, the dendritic cells

**Data availability statement:** All relevant data are within the manuscript and its Supporting Information files.

**Funding:** This work was funded through a VENI Grant (to CWEE) from the Netherlands Organization for Scientific Research (NWO-VENI 09150162010181) and an Erasmus MC Fellowship (to CHG). The funders did not play any role in the study design, data collection and analysis, decision to publish or preparation of the manuscript.

**Competing interests:** The authors have declared that no competing interests exist.

retained their ability to respond to additional stimuli. Our findings provide insight into how rabies virus interacts with a crucial part of the initial immune response and highlight the need for further research into boosting immune activation by targeting dendritic cells.

## Introduction

Rabies, despite being one of the oldest known viral diseases, continues to pose a risk to human and animal health worldwide [1]. Considered a neglected tropical disease, rabies is responsible for at least 60,000 human deaths annually, most of which occur in impoverished communities in Asia and Africa. Multiple viruses of the *Lyssavirus* genus can cause rabies in humans, but 99% of the human cases are caused by infection with rabies virus (RABV), which is most commonly transmitted via the bite of an infected dog [2]. Next to dogs, mammals such as bats, foxes, raccoons, and cats pose a potential reservoir for RABV and have been associated with occasional cases of human rabies in Europe and the Americas [2–5].

RABV infections are almost always fatal once neurological symptoms arise, due to a progressive dysfunction of the nervous system. Current post-exposure treatment comprises active and passive immunization and can reliably prevent death if administered promptly after exposure [2,6]. However, dog bite victims in rabies endemic countries frequently face challenges regarding availability or costs of the treatment, or lack of awareness [7,8]. Thus, combatting rabies remains a challenge and can only be achieved through combining different strategies as defined by the Zero by 30 plan [9].

An effective immune response is considered crucial to prevent rabies disease but appears to be lacking upon natural RABV infection. For instance, neutralizing antibodies – if any – are found only in late stages of infection [4,6], once the virus has spread throughout the brain. Despite massive virus replication in the central nervous system (CNS), histopathological changes and inflammatory signs are mild, and several immunosuppressive mechanisms employed by RABV in the CNS stage of disease have been described [6,10–13]. However, the initial interaction between RABV and the host immune system occurs much earlier, at the peripheral stage of infection [14,15]. Typically, at the peripheral entry site invading pathogens are expected to activate innate immune cells such as dendritic cells (DCs) and macrophages, which subsequently migrate to the lymphoid tissue and initiate an effective virus specific immune response with activated T and B cells [16,17]. Previous studies by our group using monocyte-derived macrophages found a lack of activation following exposure to RABV [18]. At the same time, transcriptomic analysis revealed a unique phenotype of RABV-exposed macrophages with indications of an antiviral response [19].

For the present study, we hypothesized that street RABV fails to activate monocyte-derived DCs (moDCs), thereby interfering with the onset of an adaptive immune response. MoDCs are attracted to the site of inflammation in high numbers [20,21] and are highly efficient antigen presenting cells. As such they can activate naïve T cells and stimulate their proliferation. A failure in the antiviral DC response is likely to impair the subsequent adaptive immune response as well [17]. Bacterial or viral components, such as lipopolysaccharide (LPS) or viral RNA, can activate DCs [22,23]. Upon activation, DCs upregulate maturation markers such as CD80, CD86, CD83, and HLA-DR, and migrate to secondary lymphoid organs. This goes along with the upregulation of chemokine receptor CCR7 and downregulation of CCR1, -2, and -5 [21,22,24]. Therefore, we chose to study the expression of these markers, as well as the

ability to stimulate T cell activation, of moDCs exposed to RABV to characterize a potential failure of initiating an anti-RABV immune response.

Previous studies on the interaction of RABV with DCs have been performed using different DC models and RABV strains. While some studies found DCs susceptible to RABV infection, or activated by RABV exposure, others did not [25–30]. The human DC response to RABV exposure thus remains unclear. Therefore, for the present study we chose a structured approach to carefully investigate the response of human moDCs to exposure to three different RABV strains. We compared two street RABV strains, derived from fatal human rabies cases related to a domestic dog and a silver-haired bat, respectively, and the attenuated vaccine-based strain SAD P5/88 Potsdam [31–34]. We analyzed the susceptibility of moDCs to infection as well as the effect of virus exposure on their cytokine response and maturation and migratory phenotype. In addition, the capacity of RABV-exposed moDCs to stimulate T cell activation was assessed. Lastly, we measured if exposure to RABV impaired their response to LPS. Although we observed differences between the capacity of the three RABV strains to infect and activate moDCs, overall, a fully activated antiviral moDC phenotype was not induced upon RABV exposure.

## Results

### MoDCs are permissive to a bat-associated and a vaccine-based, but not a dog-associated RABV strain

The susceptibility of moDCs to three different RABV strains was assessed. Immunofluorescence analysis showed that moDCs were susceptible to infection with the pathogenic silver-haired bat rabies virus (SHBRV) and the attenuated SAD P5/88 Potsdam (SAD P5) strains, but not the pathogenic dog-associated RABV (dogRV) (Figs 1A and S1). The immunofluorescence images clearly showed replication clusters that stained positive for the RABV nucleoprotein (RABV-N) in the cytoplasm of moDCs exposed to SHBRV and SAD P5. Co-staining for CD86, a maturation marker present on the surface of moDCs, and the lack of positive RABV-N staining in moDCs exposed to inactivated SHBRV, confirm that the observed replication clusters indeed showed intracellularly replicating virus (S1 Fig). However, only a few cells were found to be infected by either of the strains SAD P5 or SHBRV. No RABV-N staining was found in moDCs exposed to dogRV. These findings were confirmed by flow cytometric analysis using the same anti-RABV-N antibody (Fig 1B). The percentage of infected moDCs 48 h post infection (p.i.) was comparably low (< 1% at an MOI of 0.1, and between 4.8% to 10.9% at an MOI of 5) (S1 Table). Despite the low infection percentage, an increase in viral titers 48 h p.i. was observed with all three MOIs of 0.1, 1, and 5 of SHBRV and SAD P5, indicating productive infection of moDCs (Fig 1C). Viral titers in the supernatants of moDCs exposed to dogRV, in contrast, did not rise above the titer in the supernatant collected after washing off the inoculum, further supporting the evidence that moDCs are not permissive to this strain. All three RABV strains replicate efficiently in the highly susceptible neuroblastoma cell line SH-SY5Y at low MOIs. In conclusion, these results demonstrate the susceptibility and permissiveness of moDCs to SHBRV and SAD P5, but not to dogRV.

### SHBRV and SAD P5 exposure affects maturation and migration marker expression in moDCs

After assessing permissiveness of moDCs to RABV infection, the activation status of moDCs upon exposure to the different RABV strains was examined by flow cytometry. An increase in maturation marker expression was observed in moDCs exposed to SHBRV and SAD P5 at MOIs of 1 and 5 (Fig 2A, depicted in grey), but not in moDCs exposed to dogRV, which

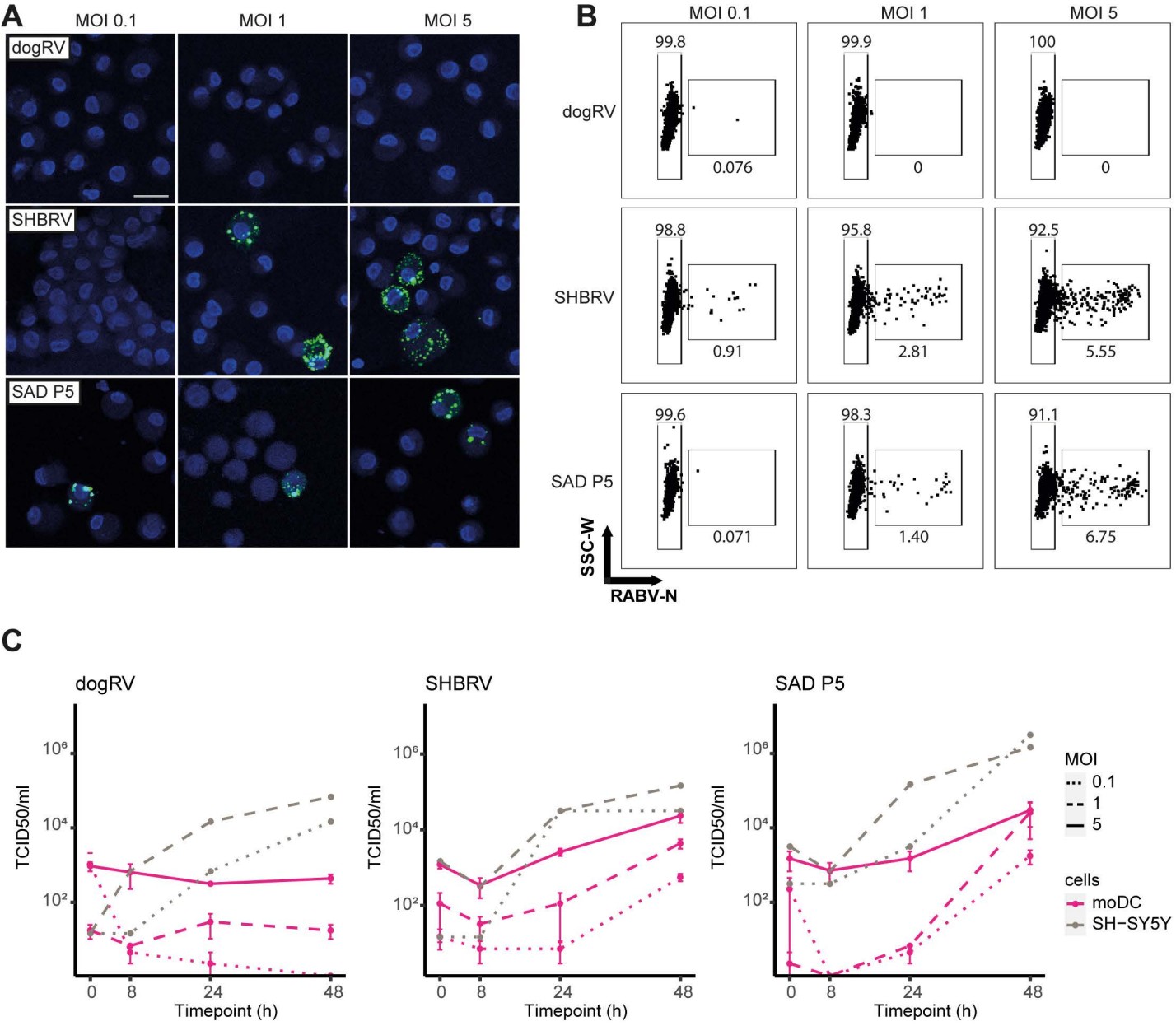

**Fig 1. Human monocyte-derived DCs are permissive to SHBRV and SAD P5, but not dogRV infection.** MoDCs were infected with RABV strains dogRV, SHBRV, and SAD P5, at MOIs of 0.1, 1, and 5 for 48 h. **A** Presence of intracellular RABV in moDCs infected with SHBRV or SAD P5 48 h p.i. Visualized by immunofluorescence imaging using anti-RABV-N antibodies (green), and Hoechst to stain nuclei (blue). Images shown for one representative donor. Scalebar represents 20 μm. **B** Infection percentages of one representative donor as measured by flow cytometry using the same anti-RABV-N antibody. **C** Increase in viral titers in supernatants of infected moDCs or SH-SY5Y cells. Data is shown as mean ± SEM. $n$ = 3 individual donors for moDCs, $n$ = 1 for SH-SY5Y cells.

correlated with their susceptibility to infection. The increase was particularly clear for the markers CD80, CD86, and HLA-DR, for moDCs exposed to SHBRV, and, to a lesser extent, for moDCs exposed to SAD P5, but only at the high MOI of 5. The co-inhibitory marker PD-L1 was also more highly expressed in moDCs exposed to SHBRV at an MOI of 5, and in moDCs exposed to SAD P5 at both MOIs 1 and 5, compared to mock-exposed moDCs. This increase was statistically significant, although it did not reach expression levels that were

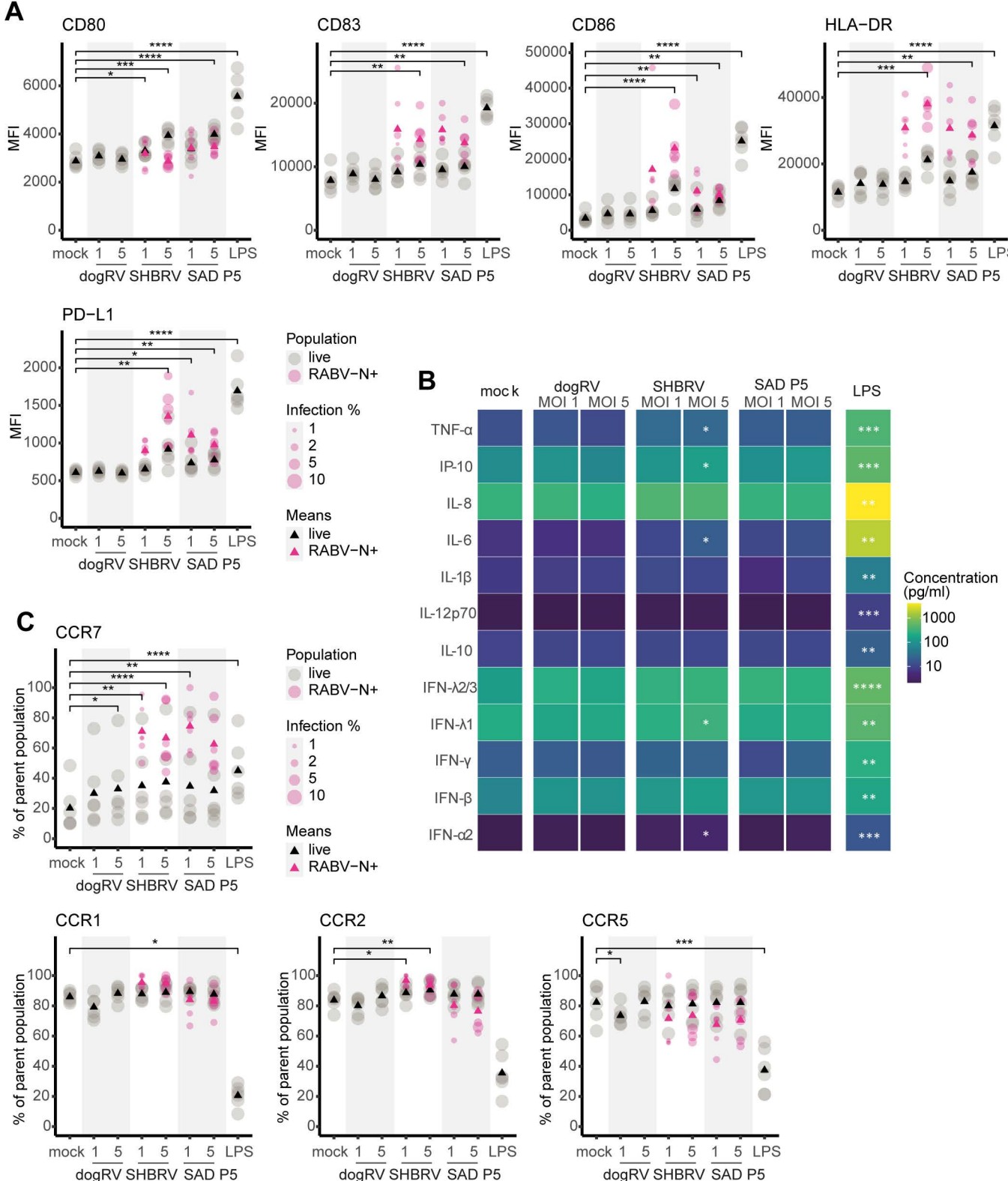

**Fig 2. moDC activation in response to RABV exposure.** MoDCs were exposed to RABV strains dogRV, SHBRV, and SAD P5, at MOIs of 1 and 5 for 48 h. **A** Activation marker expression of the complete live moDC fraction (grey), and gated on the infected (RABV-N+) moDC fraction (pink). Triangles represent the mean values of the respective population. Dot sizes of the RABV-N+ fraction correspond to the percentage of infected cells. **B** Cytokine concentrations in

moDC culture supernatants 48 h p.i. Tiles represent mean values of all donors. **C** Migration marker expression. Colors and shapes as described for (A). $n = 6$ individual donors. Only statistical comparisons of total exposed populations (grey) with mock controls are shown. MFI – median fluorescence intensity. **** $p \leq 0.0001$; *** $p \leq 0.001$; ** $p \leq 0.01$; * $p \leq 0.05$.

observed upon LPS stimulation. It should be mentioned, however, that LPS is a very strong DC stimulus and was therefore used as a positive control. It induced a similar response as the dsRNA analogue poly-I:C (S2 Fig). We chose to use LPS, since we would expect LPS as outer membrane component of gram-negative bacteria to be present in natural infections caused by animal bites or scratches. An intracellular RABV-N staining was used to distinguish between infected and non-infected moDCs (S3 Fig). Within the RABV-N$^+$ fraction, the expression levels of CD83, CD86, HLA-DR, and PD-L1 increased to higher levels than within the total live moDC fraction. However, the RABV-N$^+$ population comprised only a small fraction of the overall moDC population, as the infection percentage at a high MOI of 5 was still < 11%.

In Fig 2B it is shown that the levels of cytokine release support the small effect of RABV exposure on the overall moDC population. The panel we used includes crucial cytokines of the innate antiviral response such as type I, II, and III interferons (IFNs). Only minor increases in cytokine concentrations were measured in the supernatants of moDCs exposed to RABV compared to mock-infected moDCs. However, the increase in TNF-α, IP-10, IL-6, IFN-λ1, IFN-β, and IFN-α2 concentrations was statistically significant for exposure to SHBRV at an MOI 5, compared to the mock controls. Although this was still considerably lower than the cytokine levels reached upon LPS stimulation, it was measurable despite the low percentage of infected moDCs.

Looking at migration markers (Fig 2C), at 48 h p.i. a statistically significant increase in CCR7 expression was observed for all RABV strains compared to the mock controls, although there was a high between-donor variability. CCR7 expression within the small RABV-N$^+$ fraction was even higher than in the LPS-stimulated positive controls. On the other hand, the expression levels of CCR1, CCR2, and CCR5 of RABV-exposed moDCs differed only marginally from those of mock-exposed moDCs. This was the case for both the overall moDC populations and the RABV-N$^+$ fractions. It is of note that the expression of these markers is downregulated upon activation, as seen clearly for the LPS-stimulated moDCs. Thus, a clear, but moderate, effect of RABV exposure on the migration marker expression was found only for CCR7.

## RABV-exposed moDCs do not stimulate naïve T cell activation

To gain further insight into whether RABV-exposed moDCs could induce an adaptive immune response, RABV-exposed moDCs were co-cultured with autologous naïve T cells. T cell activation was measured by quantifying the cytokine levels in co-culture supernatants and by measuring T cell proliferation. Average concentrations of most cytokines were significantly higher in supernatants of T cells co-cultured with LPS-stimulated moDCs compared to mock-exposed moDCs (Fig 3C). No specific pattern, however, was observed for the cytokine release in co-cultures of T cells with RABV-exposed moDCs: some cytokines showed lower concentrations compared to the mock controls (such as TNF-α, IFN-β, IL-10), and some higher (IP-10, IL-6, IFN-λ1, GM-CSF). This was particularly the case for T cells co-cultured with SHBRV-exposed moDCs. T cell proliferation was measured by quantification of carboxyfluorescein succinimidyl ester (CFSE) signal by flow cytometry, in combination with the side scatter (Fig 3A). In general, the percentage of proliferated T cells co-cultured with moDCs, RABV-exposed or mock-exposed, was higher than that of unstimulated T cells (Fig 3B). However, moDCs exposed to RABV did not stimulate T cell proliferation to a greater extent than mock-exposed moDCs. The mean fraction of proliferated T cells was around 5% for these conditions, compared to a significantly higher fraction of 12.5% proliferated T cells

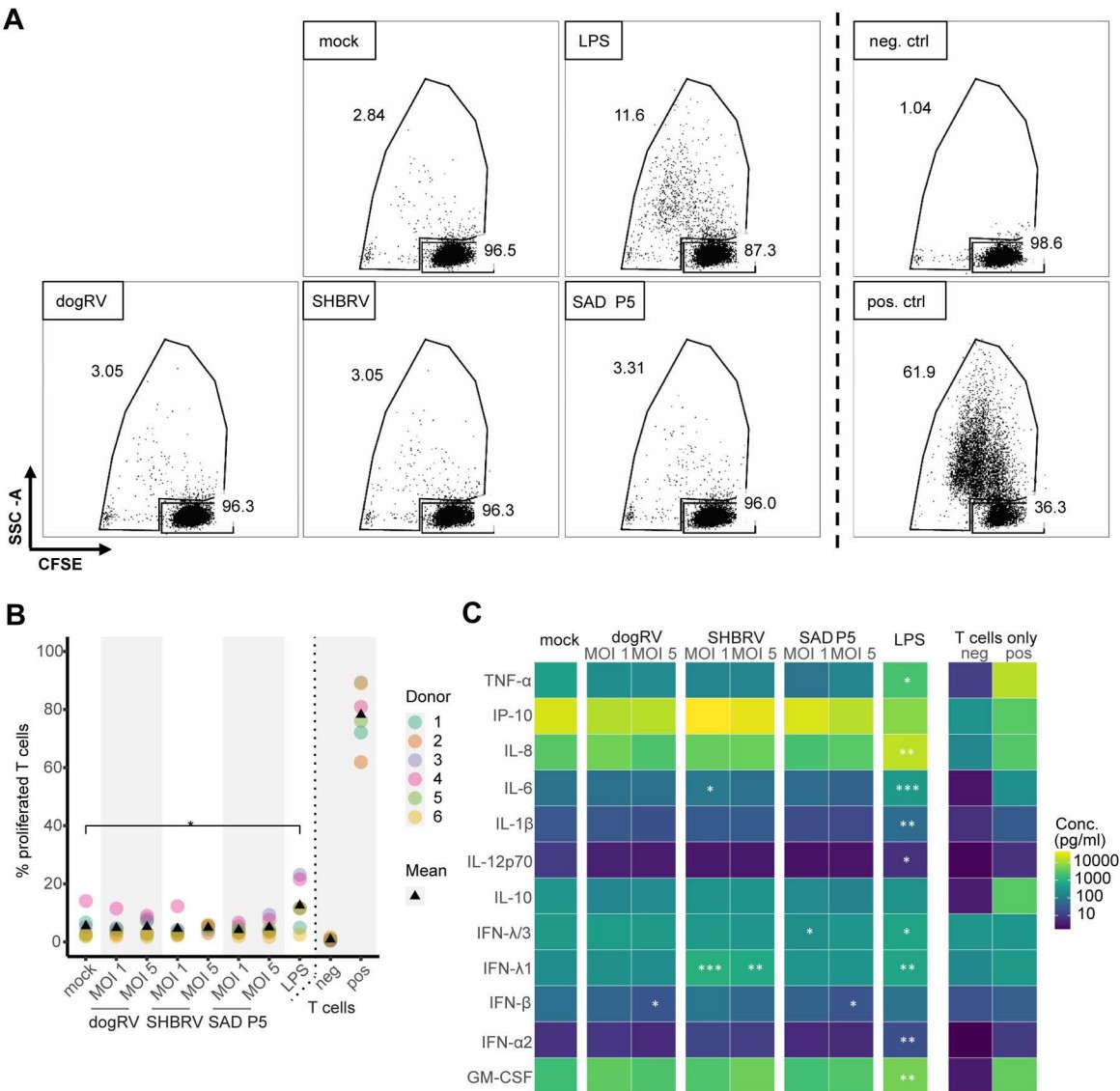

**Fig 3. T cell proliferation and activation following co-culture with RABV-exposed moDCs.** MoDCs were exposed to RABV strains dogRV, SHBRV, and SAD P5, at MOIs of 1 and 5, or stimulated with 10 ng/mL of LPS, for 48 h. CFSE-labeled autologous T cells were added at a ratio of 10:1 for 2.5 days. As negative (neg) and positive (pos) controls, T cells were cultured without moDCs, with or without CD3/CD28 activation beads. Dashed lines separate the results for co-cultures with moDCs exposed to the different conditions (left) and cultures of only T cells as controls (right). **A** Proliferated and non-proliferated T cell populations of one representative donor (donor 2), as determined by measurement of sideward scatter (SSC) and CFSE fluorescence intensities. **B** Percentage of proliferated T cells for all donors. Colors represent the different donors and triangles show mean values. **C** Cytokine concentrations in co-culture supernatants. Tiles represent mean values of all donors. T cell only conditions were excluded from statistical analysis. $n = 6$ individual donors. Only statistical comparisons with mock controls are shown. *** $p \leq 0.001$; ** $p \leq 0.01$; * $p \leq 0.05$.

co-cultured with LPS-stimulated moDCs. This supports the observation that RABV exposure of moDCs does not enhance activation of naïve T cells in *in vitro* co-cultures.

## Exposure to RABV does not suppress the LPS response of moDCs

Ultimately, we assessed the effect of RABV exposure on the LPS response of moDCs. 24 h after exposure to one of the three RABV strains, moDCs were stimulated with LPS for 48 h,

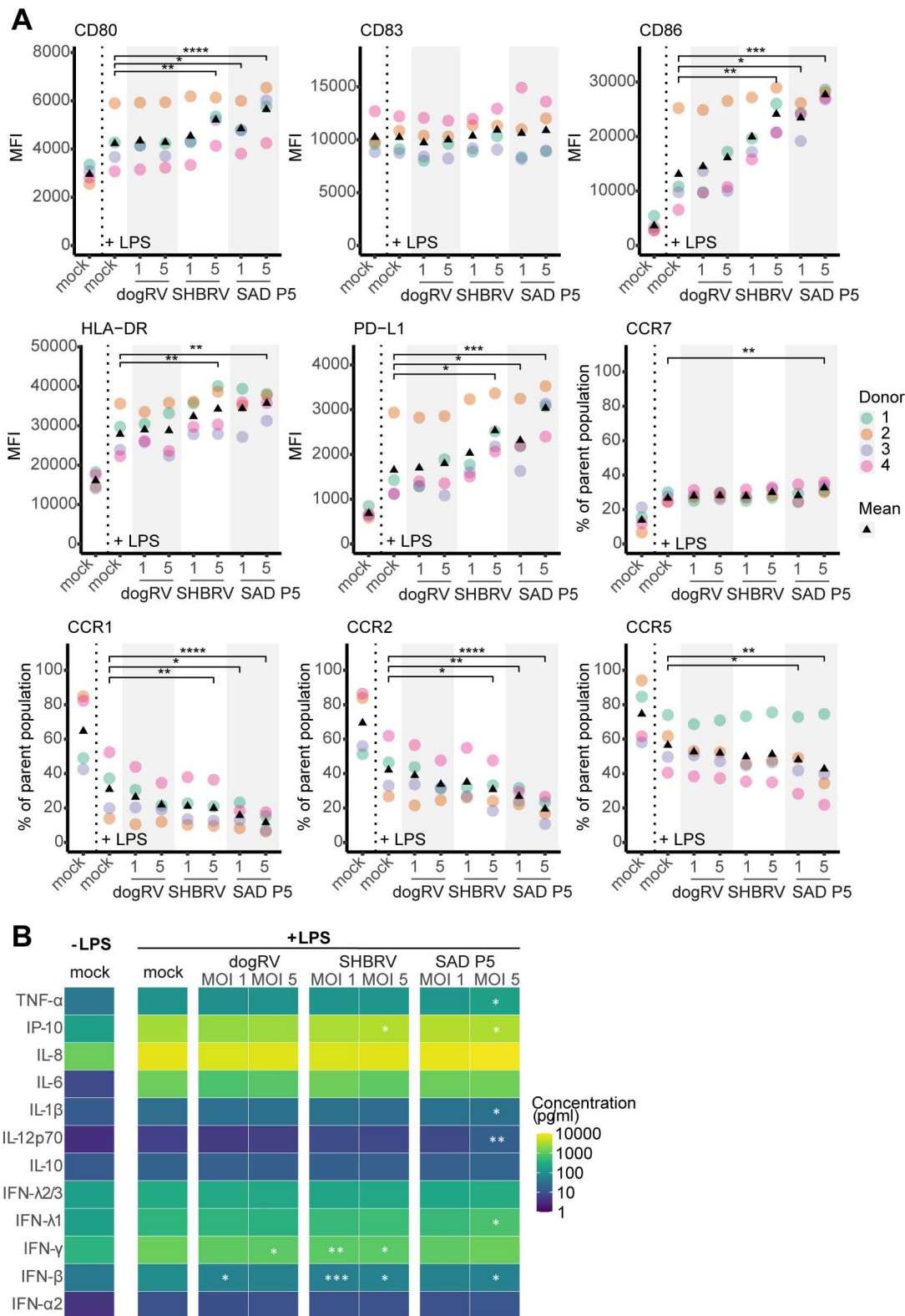

**Fig 4. LPS response of moDCs exposed to RABV.** moDCs were exposed to RABV strains dogRV, SHBRV, and SAD P5, at MOIs of 1 and 5, treated with LPS at 24h and analyzed 48h post-stimulation. **A** Activation and migration marker expression. Dot colors represent different donors and triangles show mean values. **B** Cytokine concentrations in moDC culture supernatants.

Tiles represent mean values of all donors. $n$ = 4 individual donors. Only statistical comparisons with LPS-stimulated mock controls are shown. MFI – median fluorescence intensity. **** $p \leq 0.0001$; *** $p \leq 0.001$; ** $p \leq 0.01$; * $p \leq 0.05$.

after which the levels of activation and migration markers, as well as released cytokines, were measured. Fig 4 shows that RABV exposure did not affect the response of moDCs to LPS. In fact, moDCs exposed to high MOIs of SHBRV and SAD P5 seemed to show an even more activated phenotype following LPS stimulation than mock-infected moDCs (Fig 4A). Looking at the infected RABV-N+ moDC population, this activation was even more enhanced (S4 Fig). This was observed for both maturation markers (CD80, CD86, HLA-DR, PD-L1), and migration markers (CCR7, CCR1, CCR2, CCR5). Interestingly, this was not the case for the dogRV-exposed moDCs, which showed an LPS response comparable to that of mock-infected moDCs. This was further supported by analysis of cytokine concentrations in moDC supernatants (Fig 4B). Generally, cytokine levels in the supernatants of LPS-stimulated RABV-exposed resembled those of LPS-stimulated mock-exposed moDCs. Minor, but significantly higher concentrations were found for some cytokines in the supernatants of moDCs exposed to SAD P5 at an MOI of 5. The concentrations of IFN-γ and IFN-β were significantly lower in the supernatants of LPS-stimulated SHBRV-exposed compared to LPS-stimulated mock-exposed moDCs, but still clearly higher than for the unstimulated moDCs. These results show that exposure to RABV does not suppress the LPS response of moDCs. In contrast, exposure to SHBRV and SAD P5 at a high MOI of 5, but not dogRV, seems to enhance their activation following LPS stimulation.

## Discussion

In this study, we focused on a structured and comprehensive comparison of the human moDC response to exposure to three different RABV strains. We investigated the susceptibility, cytokine response, maturation, migration, and LPS response of RABV-exposed moDCs, as well as their capacity to stimulate T cell activation. Our results reveal marginal differences in the susceptibility and antiviral response, depending on the strain. We did not observe infection or activation of moDCs upon exposure to dogRV. In contrast, human moDCs were permissive to infection by SHBRV and SAD P5, although the infection percentage was low and the contribution of these infected cell populations to the overall moDC activation is presumably small. Exposed moDCs were generally more activated at high MOIs, which was most pronounced for the infected moDC population. The overall activation of moDCs was not sufficient to result in substantially increased cytokine levels or T cell activation. Nevertheless, RABV-exposed moDCs retained their ability to respond to LPS stimulation. This study gives an overview of the (in)ability of human moDC to respond to RABV exposure, by using different RABV strains and looking at different aspects of the immune response.

Effective antigen presentation is crucial for the onset of an adequate B and T cell response to virus infection, and requires uptake, processing, and presentation of viral antigens by DCs. Their activation in response to different viruses is well known. For instance, DCs exposed to dengue virus upregulate maturation markers and secrete cytokines such as IFN-α and TNF [35–37]. Similar findings have been observed for influenza A virus, in which DC maturation and induction of antiviral response pathways are associated with effective viral clearance, while an impaired DC response has been associated with an increased disease severity [35,38,39]. Of note, DCs are readily infected with dengue virus, which has been associated with their strong antiviral response [35]. An active infection of moDCs likely enhances antigen uptake and may amplify moDC maturation and antigen presentation [17,29]. However, suppression or activation of moDCs does not require productive infection. moDCs can

mount a strong antiviral response to viruses like Influenza A virus or SARS-CoV-2 in absence of productive replication [40,41]. At the same time, viral exposure can suppress the moDC response despite abortive infection, as observed for vaccinia virus [42]. In previous studies by our group, we observed a suppressed phenotype of monocyte-derived macrophages without productive infection, encouraging us to study the moDC response as well [18]. In the present study, the head-to-head comparison of two street RABV strains (SHBRV and dogRV) with an attenuated RABV strain (SAD P5) showed that moDCs were permissive to both SHBRV and SAD P5, but not dogRV infection. The percentage of moDCs infected with SAD P5 and SHBRV in our study did not exceed 11%, even at a high MOI of 5. A direct comparison with other studies is difficult because results can differ based on the selected DC models, RABV strains, and methods used to assess infection. As an example, a study by Lawrence *et al.* [25] showed that the majority of murine bone marrow-derived DCs (BMDCs) exposed to the attenuated RABV vaccine strain SPBN at an MOI of 10 stained positive for intracellular RABV-N, whereas Senba *et al.* and Faul *et al.* were able to show that RABV strains SPBN, CVS, and ERA could enter, but not replicate in the murine DC cell line JAWS II at an MOI of 10 [26,27]. While these studies used murine DC models, we chose the human moDC model and found productive infection in two out of the three RABV strains. A potential mechanism for the lack of susceptibility of moDCs to dogRV, in contrast to SHBRV and SAD P5, could be differences in glycoprotein (G protein) expression. The G protein plays a crucial role for RABV infectivity and pathogenicity and has been associated with reduced binding of wild-type compared to attenuated RABV strains and subsequent DC maturation [29,30,43]. Thus, potential differences in the G protein of the strains used in our study may also account for the differences observed in their ability to infect and activate moDCs.

The observation that dogRV, which did not productively infect moDCs, failed to induce moDC maturation in our model suggests that maturation may require active viral replication. Correspondingly, Yang *et al.* found that the extent of DC activation correlated with the level of intracellular RABV leader RNA [29], which the cytosolic sensor RIG-I recognizes to initiate a type I interferon response [27,44]. Since the leader RNA is transcribed within the host cell during the viral replication cycle [45], the lack of activation observed with dogRV may be related to a lack of synthesized leader RNA. Interestingly, the small fraction of moDCs productively infected with SAD P5 and SHBRV displayed a clearly activated phenotype, supporting the hypothesis that active viral replication leads to the host cell's activation [25,29].

Since we observed a slight activation of moDCs upon SHBRV, but not dogRV exposure, we are intrigued whether transcriptomic analyses can reveal further differences between these two street RABV strains and complete the picture of the anti-RABV moDC response. We previously observed a unique transcriptomic profile of monocyte-derived macrophages upon SHBRV exposure [19] and hypothesize that this might be the case for moDCs as well, and moreover, can give further insight into the activation state of the infected compared to the uninfected moDC fractions.

Based on previous studies, we hypothesized that the attenuated RABV strain would be capable of activating moDCs, while street RABV strains would not [46–48]. For example, Senba *et al.* found a strongly enhanced JAWSII cell activation in response to ERA compared to CVS infection, as measured by maturation marker expression and type I IFN production. They used the low-pathogenic SAD P5 ancestral strain ERA, as well as the pathogenic fixed strain CVS, to infect JAWS II cells [26]. Similar results were obtained by Yang *et al.* using murine BMDCs and the strains CVS-B2c (attenuated) and DRV-Mexico (street RABV, isolated from a rabid dog) [29]. In contrast, in our study using human moDCs, we found comparable results between the attenuated SAD P5 and the pathogenic SHBRV strain, emphasizing that the extent of moDC activation may be highly dependent on the model and the RABV

strains used. The strain used is this study, SAD P5, is derived from the ancestor strain "Street Alabama Dufferin", isolated 1935 from a rabid dog, through serial passaging and plaque purification, and has been successfully used for oral vaccination of foxes in Europe [33]. The strain reliably induces the production of virus neutralizing antibodies in raccoon dogs and foxes [49,50], but to our knowledge its immunogenicity has not been characterized in detail. In the present study, moDC maturation in response to SAD P5 was only minor and comparable to that of the street RABV strain SHBRV. In addition to surface maturation marker expression, we quantified the cytokine release of RABV-exposed moDCs as a measure of moDC activation. Our panel included characteristic cytokines of the innate antiviral response, such as type I IFNs, IL-6, and TNF-α. Our results indicate that moDCs exposed to SHBRV, but not to SAD P5 or dogRV, indeed show signs of an innate antiviral response when exposed to a high MOI. However, the observed response seems to be insufficient to induce full moDC activation.

None of the RABV-exposed moDCs exhibited a clear migratory phenotype. Migration to the lymph nodes is an important factor for the initiation of an adaptive immune response, since this is where DCs come in contact with T cells [22]. CCR7 expression is characteristic for DC homing to lymph nodes [22,51]. Expression of CCR1, CCR2, and CCR5, on the other hand, directs immune cells to peripheral sites of inflammation, and is downregulated upon LPS stimulation [21,24,52]. In a study comparing different respiratory viruses, an upregulation in CCR7 expression, and downregulation in CCR1, CCR2, and CCR5 expression, was observed on moDCs exposed to influenza A virus, but not human metapneumovirus or human respiratory syncytial virus. This correlated with the strains' ability to induce effective or incomplete immunity, respectively [53]. In our study, RABV exposure induced moderate upregulation of CCR7 expression on moDCs, which was more pronounced within the infected moDC fractions. However, no pronounced differences in the expression of CCR1, CCR2, and CCR5 were observed between mock- and RABV-exposed moDCs. This indicates that the migratory capacity of RABV-exposed moDCs might be impaired and requires further investigation.

In the secondary lymphoid organs, activated moDCs stimulate T cell proliferation, whereas tolerogenic moDCs have an inhibitory effect [22,54]. In our study, T cells were not activated to a greater – or lower – extent upon co-culture with RABV- compared to mock-exposed moDCs. We chose this approach of co-culturing RABV-exposed moDCs with a naïve autologous T cell population, since this is expected to mimic the interaction in the lymph nodes of previously unexposed individuals. However, this approach disregards potential co-stimulatory effects that might be present and necessary in natural infections to induce efficient T cell priming. While our results support that there's an insufficient interaction of moDCs with T cells, further studies are needed to fully comprehend their contribution to the lack of adaptive immune responses in RABV infections. Effective T cell priming requires not only moDC maturation and migration – which seemed to be the case to a certain extent in response to SHBRV and SAD P5 exposure – but is the result of complex signaling processes. Other aspects, such as antigen processing and cross-presentation, might be inhibited by RABV exposure. Interestingly, the infected moDC fractions displayed a more activated phenotype than the uninfected fractions, suggesting that they might be more capable of stimulating T cell activation. However, this would not be representative of the natural situation due to the low susceptibility of moDCs.

Since a lack of moDC activation might be a result of active suppression mechanisms employed by RABV, we investigated the ability of RABV-exposed moDCs to respond to LPS. A common viral strategy of suppressing DC activation has been previously shown for Ebola virus, SARS CoV-2 and herpes simplex virus [55–59]. A lack of responsiveness of DCs to LPS stimulation was demonstrated following pre-exposure to SARS CoV-2 S protein, herpes

simplex virus $\gamma_1$34.5 protein, or Ebola virus VP35 protein [56,59,60]. It was not observed for RABV in our experiments. Upon LPS stimulation, RABV-exposed moDCs showed an increase in maturation marker expression similar to that of mock-exposed moDCs. Moreover, moDCs previously exposed to SHBRV and SAD P5 were even more activated than the mock-exposed moDCs, suggesting that the antiviral immune response can be restored upon additional moDC stimulation. In natural infections, this additional stimulus might already be present due to the tissue damage and influx of bacteria accompanying the dog bite. Hence, this calls for the investigation of the moDC activation state under these natural circumstances.

For the present study, we chose to look at moDCs, since these are among the immune cell types most likely to encounter the virus at the initial site of exposure in natural rabies infections. Monocytes are recruited to the wound, where they differentiate into DCs. However, there are other DC subtypes that might be relevant for the antiviral response and are worth investigating, such as conventional DCs 1 and 2 and plasmacytoid DCs [20,21].

Altogether, we observed insufficient moDC activation upon RABV exposure, with slight differences between RABV strains. The most remarkable difference was the lack of susceptibility of moDCs to dogRV, in contrast to SHBRV and SAD P5. While no signs of an antiviral response of moDCs were observed to exposure of dogRV, moDCs exposed to SHBRV and SAD P5 at high MOIs showed a slightly more mature phenotype, yet not sufficient to stimulate T cell activation. Taken together, these findings support the hypothesis that RABV fails to sufficiently activate moDCs, which may contribute to the lack of an adaptive immune response that is seen in natural rabies virus infections. Further studies can provide insight into the failure of RABV-exposed moDCs to stimulate T cell activation, as well as the reasons behind the varying susceptibility and responses of moDCs to different RABV strains.

## Materials and methods

### Cells and viruses

Three different rabies virus (RABV) strains were used for this study. The attenuated vaccine-based strain SAD P5/88 Potsdam, isolate 9509TCH, was acquired from Institute Pasteur, Paris, France through the European Virus Archive Global (EVAg, Ref-SKU: 014V-01931). The highly pathogenic silver-haired bat rabies virus (SHBRV) strain was originally obtained from the brain of a fatal bat-acquired human case of rabies in North America, and is commonly used as street RABV strain in experimental infections [34,61]. The dogRV strain was isolated from the medulla oblongata of a Nepali patient who developed fatal rabies disease after a dog bite, as described previously [31,32].

All viruses were propagated on the highly susceptible human neuroblastoma cell line SH-SY5Y in Advanced DMEM/F12 (Gibco) supplemented with 2% (v/v) Fetal Calf Serum (FCS) (Sigma-Aldrich), 1x Penicillin/Streptomycin (Capricorn Scientific), 2 mM L-glutamine (Capricorn Scientific), 1x non-essential amino acids (Capricorn Scientific), and 0.8 mg/mL sodium bicarbonate (Gibco). Cells were maintained at 37 °C and 5% $CO_2$ atmosphere. Viral titers were determined using the endpoint dilution method as described below. Inactivated RABV (iRV) was obtained by incubation of SHBRV virus stocks with β-Propionolactone (Acros Organics) 1:4000 and 1.5 mg/mL sodium bicarbonate for 2 d, followed by 6 h incubation at 37 °C.

### Peripheral blood mononuclear cell isolation and dendritic cell differentiation

Peripheral blood mononuclear cells (PBMCs) were isolated from buffy coats by density gradient centrifugation using Ficoll Paque PLUS (GE Healthcare). Buffy coats were obtained from

non-rabies vaccinated and non-smoking healthy blood donors (Sanquin). Written informed consent for use of donated blood for research purposes was obtained by the Sanquin blood bank. The CD14+ and CD3+ cell fractions were isolated from PBMCs using magnetic activated cell sorting according to manufacturer's guidelines (Miltenyi Biotec) to obtain monocytes and T cells, respectively. Sorting purity was assessed by flow cytometry and was found to be > 92%.

To obtain monocyte-derived dendritic cells (moDCs), monocytes were seeded at a density of 200,000 cells per well of a 96-well flat-bottomed plate and differentiated in RPMI-1640 medium (Capricorn Scientific), supplemented with 10% (v/v) FCS, 1x Penicillin/Streptomycin, 1x GlutaMAX (Gibco), 20 ng/mL Granulocyte-Macrophage Colony Stimulating Factor (GM-CSF, Miltenyi Biotec), and 20 ng/mL IL-4 (Peprotech), for 6 days. Cells were cultured at 37 °C and 5% $CO_2$ and medium was replaced on day 4.

### Infection of monocyte-derived dendritic cells

On day 6 of differentiation, moDCs were harvested by gentle pipetting and rinsing the wells with cold PBS-5 mM EDTA. moDCs were spun down at 300 xg for 5 min and resuspended in serum-free medium. RABV dilutions were added at an MOI of 0.1, 1, or 5, for 1 h at 37 °C with gentle mixing of the cells every 10 min. Mock-infected moDCs were incubated with SH-SY5Y cell conditioned medium. moDCs were spun down, resuspended in complete medium, and seeded at a density of 100,000 cells per well in 96-well flat-bottomed plates. For susceptibility experiments ($n = 3$ donors), an additional washing step was included. 0 h time-point samples were taken immediately after the cells were washed and seeded in the plates. moDCs were harvested for immunohistochemistry and flow cytometry analysis 48 h post-infection. As positive control, moDCs were incubated with 10 ng/mL of LPS for 48 h. poly-I:C was used at a concentration of 10 μg/ml. A total of $n = 6$ donors were used to assess maturation and surface marker expression, as well as cytokine release, following exposure to RABV at MOIs of 1 or 5, or LPS stimulation. To determine the response of RABV-exposed moDCs ($n = 4$ donors) to LPS stimulation, 10 ng/mL of LPS were added 24 h post-infection for another 48 h, after which moDCs were harvested and stained for maturation and migration markers (see below).

### T cell co-culture

CD3+ T cells ($n = 6$ donors) were labeled with 2 μM carboxyfluorescein succinimidyl ester (CFSE) as described previously [18], and added to autologous moDCs 48 h post-infection at a ratio of 10:1. moDC supernatants were removed before addition of T cells. moDCs and T cells were co-cultured in RPMI-1640 supplemented with 10% (v/v) pooled human serum (Sanquin), 1x Penicillin/Streptomycin, and 1x GlutaMAX at 37 °C and 5% $CO_2$. As controls, non-stimulated T cells and T cells stimulated with 2 μl of Dynabeads Human T-Activator CD3/CD28 (Gibco) were taken along. After 2.5 d of co-culture, T cells were harvested, and proliferation was determined by flow cytometry. In addition, supernatants were taken for quantification of cytokine release.

### $TCID_{50}$ endpoint dilution assay

For quantification of infectious virus in moDC supernatants, moDCs of $n = 3$ individual donors were exposed to the three RABV strains at MOIs of 0.1, 1, and 5. As controls, SH-SY5Y cells were infected at MOIs of 0.1 and 1. Supernatants were harvested at 0 h, 8 h, 24 h, and 48 h post-infection. Viral titers were quantified using the median tissue culture infective dose ($TCID_{50}$) endpoint dilution method of Reed and Muench [62]. SH-SY5Y cells were seeded at a density of 4.5 x $10^4$ cells per well in 96-well plates 1 day prior to sample addition.

Samples were added in triplicates in 10-fold dilution series in serum-free medium for 1 h at 37 °C, after which medium was replaced with complete medium. After 3 days of incubation at 37 °C and 5% $CO_2$, cells were fixed with 80% Acetone for 30 min. Cells were incubated with FITC-conjugated RABV-N antibody (Fujirebio) for 1 h at 37 °C, and infection was scored using a Zeiss Axiovert A1 inverted microscope with Colibri 7 light source for fluorescence imaging.

## Flow cytometry

moDCs or T cells were harvested by gentle pipetting and rinsing with cold PBS-5mM EDTA. Live/dead staining was performed using the fixable viability dye ZombieViolet (BioLegend) for 30 min at RT. Fc receptors were blocked by 5 min incubation with Human TruStain FcX blocking reagent (BioLegend) at RT, followed by surface staining for 30 min at 4 °C. For maturation marker staining, anti-CD83-PE-Cy7 (clone HB15e), anti-CD80-PerCP-Cy5.5 (clone 2D10), anti-HLA-DR-APC-Cy7 (clone L243), anti-CD86-AF647 (clone IT2.2), and anti-PD-L1-BV785 (clone 29E.2A3, all BioLegend) were used. For migration marker staining, anti-CCR1-PE-Cy7 (clone 5F10B29), anti-CCR2-APC-Cy7 (clone K036C2), anti-CCR5-BV785 (clone J418F1), and anti-CCR7-BV711 (clone G043H7, all BioLegend) were used. T cells were stained using anti-CD3-PerCP (clone HIT3a), anti-CD4-BV711 (clone SK3), and anti-CD8-BV605 (clone SK1, all BioLegend). For quality control of magnetic activated cell sorting efficacy, anti-CD14-PE (clone MφP9, BD Biosciences) and anti-CD3-PerCP (clone HIT3a, BioLegend) were used. Following surface staining, cells were fixed and permeabilized using the Cytofix/Cytoperm kit (BD Biosciences). Intracellular staining was performed for 30 min at 4 °C using a FITC-conjugated RABV-N antibody (Fujirebio). Fluorescence intensities were measured with the BD FACSLyric flow cytometer (BD Biosciences), and data were analyzed using FlowJo V10.8.1.

## Immunofluorescence imaging

moDCs were harvested, blocked with Fc receptor blocking solution, and fixed/permeabilized as described above. Intracellular staining was performed for 30 min at 37 °C using FITC-RABV-N (Fujirebio) and anti-CD86-AF647 (clone IT2.2, BioLegend) antibodies, and Hoechst 33342 (Sigma Aldrich) for nuclei counterstaining. Cells were embedded with the ProLong Diamond Antifade Mountant (Invitrogen) and imaged using a Zeiss LSM 700 confocal laser scanning microscope. Images were processed using FIJI ImageJ.

## Multiplex cytokine quantification

Supernatants for quantification of cytokine release by moDCs ($n$ = 6 donors) were taken 24 h and 48 h post-infection. For experiments with LPS stimulation following RABV exposure, supernatants of moDCs ($n$ = 4 donors) were taken before LPS stimulation (*i.e.*, 24 h post-infection), 24 h, and 48 h after LPS stimulation. Supernatants of T cells co-cultured with autologous moDCs ($n$ = 6 donors) were taken after 2.5 days of culture. Cytokine levels were quantified using the LEGENDplex Human Anti-Virus Response 13-plex bead-based assay kit (BioLegend) according to manufacturer's instructions. Beads were measured with the BD FACSLyric flow cytometer and data were analyzed using the LEGENDplex Data Analysis Software Suite (BioLegend). Values outside limit of detection were extrapolated on the basis of the standard curves.

## Statistical analysis

Graphical and statistical analysis was performed using R Studio. Means between groups were compared by Friedman test and pairwise comparisons using Eisinga, Heskes, Pelzer & Te

Grotenhuis all-pairs test with exact p-values for a two-way balanced complete block design [63]. Only comparisons with mock controls are shown in figures. P values of < 0.05 were considered statistically significant.

## Supporting information

**S1 Fig. Susceptibility of moDCs to RABV strains dogRV, SHBRV, and SAD P5/88.** Immunofluorescence imaging of moDCs of $n = 3$ donors exposed to conditioned medium (mock), LPS, BPL-inactivated RABV (iRV), or either of the RABV strains dogRV, SHBRV, or SAD P5, at MOIs of 0.1, 1, and 5. moDCs were harvested 48 h p.i., fixed, and stained for intracellular RABV-N (green), CD86 (red), and nuclei (blue). Scalebar represents 20 μm.
(TIF)

**S2 Fig. moDC response to different concentrations of LPS and poly-I:C. A** Maturation and migration marker expression of moDCs in response to LPS in concentrations ranging from 0.01 ng/ml to 1000 ng/ml. **B** Maturation and migration marker expression of moDCs in response to stimulation with LPS (10 ng/ml) or poly-I:C (10 μg/ml). MFI – median fluorescence intensity.
(TIF)

**S3 Fig. Gating strategy for expression of activation and migration markers.** Surface activation and migration marker expression was measured by flow cytometry both within the overall live moDC fraction (grey) and the RABV-N$^+$ fraction (pink).
(TIF)

**S4 Fig. Maturation and migration marker expression of RABV-exposed total moDCs and RABV-N$^+$ moDCs in response to LPS treatment.** moDCs were exposed to RABV strains dogRV, SHBRV, and SAD P5, at MOIs of 1 and 5, treated with LPS at 24 h and analyzed 48 h post-treatment. Total live moDC fraction is depicted in grey and the infected (RABV-N$^+$) fraction is depicted in pink. Triangles represent the mean values of the respective population. Dot sizes of the RABV-N$^+$ fraction correspond to the percentage of infected cells. $n = 4$ individual donors. MFI – median fluorescence intensity.
(TIF)

**S1 Table. Infection percentages of moDCs exposed to SHBRV and SAD P5 RABV strains.** Measured by flow cytometry using intracellular staining of RABV-N protein.
(DOCX)

**S1 Data. Raw data of viral titers, surface expression levels, and cytokine concentrations.**
(XLSX)

## Acknowledgments

We would like to thank Institut Pasteur, Paris, for providing the SAD P5/88 Potsdam RABV strain through the EVAg repository. Elmoubashar Abu Baker Abd Farag is acknowledged for providing the tissues from which the dogRV isolate was cultured, and the Viroscience Diagnostics unit for their assistance in virus culturing and titer quantification. We would like to thank David van de Vijver for assistance with statistical analysis.

## Author contributions

**Conceptualization:** Keshia Kroh, Corine H. GeurtsvanKessel, Carmen W.E. Embregts.

**Formal analysis:** Keshia Kroh.

**Funding acquisition:** Corine H. GeurtsvanKessel, Carmen W.E. Embregts.

**Investigation:** Keshia Kroh, Merel R. te Marvelde, Lars W. van Greuningen, Brigitta M. Laksono.

**Methodology:** Corine H. GeurtsvanKessel, Carmen W.E. Embregts.

**Supervision:** Corine H. GeurtsvanKessel, Carmen W.E. Embregts.

**Visualization:** Keshia Kroh.

**Writing – original draft:** Keshia Kroh, Corine H. GeurtsvanKessel, Carmen W.E. Embregts.

**Writing – review & editing:** Keshia Kroh, Merel R. te Marvelde, Lars W. van Greuningen, Brigitta M. Laksono, Marion P.G. Koopmans, Thijs Kuiken, Corine H. GeurtsvanKessel, Carmen W.E. Embregts.

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
