## [Decision Letter · Decision Letter 0]

4 Nov 2024

PNTD-D-24-01223A comparative analysis of the dendritic cell response upon exposure to different rabies virus strainsPLOS Neglected Tropical Diseases Dear Dr. Embregts, Thank you for submitting your manuscript to PLOS Neglected Tropical Diseases. After careful consideration, we feel that it has merit but does not fully meet PLOS Neglected Tropical Diseases's publication criteria as it currently stands. Therefore, we invite you to submit a revised version of the manuscript that addresses the points raised during the review process. Please submit your revised manuscript within 60 days Jan 03 2025 11:59PM. If you will need more time than this to complete your revisions, please reply to this message or contact the journal office at plosntds@plos.org. Please include the following items when submitting your revised manuscript:* A rebuttal letter that responds to each point raised by the editor and reviewer(s). You should upload this letter as a separate file labeled 'Response to Reviewers '. This file does not need to include responses to any formatting updates and technical items listed in the 'Journal Requirements' section below.* A marked-up copy of your manuscript that highlights changes made to the original version. You should upload this as a separate file labeled 'Revised Manuscript with Track Changes '.* An unmarked version of your revised paper without tracked changes. You should upload this as a separate file labeled 'Manuscript '. If you would like to make changes to your financial disclosure, competing interests statement, or data availability statement, please make these updates within the submission form at the time of resubmission. Guidelines for resubmitting your figure files are available below the reviewer comments at the end of this letter. We look forward to receiving your revised manuscript. Kind regards, Husain PoonawalaAcademic EditorPLOS Neglected Tropical Diseases Elvina ViennetSection EditorPLOS Neglected Tropical Diseases

Shaden Kamhawi

co-Editor-in-Chief

Paul Brindley

co-Editor-in-Chief

 **Journal Requirements:** **Additional Editor Comments (if provided):****Reviewers' Comments:** Reviewer's Responses to Questions

**Key Review Criteria Required for Acceptance?**

**Methods**

-Are the objectives of the study clearly articulated with a clear testable hypothesis stated?

-Is the study design appropriate to address the stated objectives?

-Is the population clearly described and appropriate for the hypothesis being tested?

-Is the sample size sufficient to ensure adequate power to address the hypothesis being tested?

-Were correct statistical analysis used to support conclusions?

-Are there concerns about ethical or regulatory requirements being met?

Reviewer #1: (No Response)

Reviewer #2: See main review

Reviewer #3: Objectives are clear and methodology is adequate but does not allow support of the conclusions (see summary below)

**Results**

-Does the analysis presented match the analysis plan?

-Are the results clearly and completely presented?

-Are the figures (Tables, Images) of sufficient quality for clarity?

Reviewer #1: (No Response)

Reviewer #2: See main review

Reviewer #3: Results are adequately presented. Suggestions to enhance clarity are proposed in the summary below

**Conclusions**

-Are the conclusions supported by the data presented?

-Are the limitations of analysis clearly described?

-Do the authors discuss how these data can be helpful to advance our understanding of the topic under study?

-Is public health relevance addressed?

Reviewer #1: (No Response)

Reviewer #2: The paper by Kroh examines susceptibility of human monocyte-derived dendritic cells (mDCs) to infection by a group of rabies virus (RV) strains with the objective to investigate possible viral immune suppression/evasion strategies, eg. via interference with adaptive immunity.

The paper addresses a very important and meritorious question, as virus:host DC interactions are pivotal determinants of viral pathogenic strategies and host antiviral immunity. The studies are generally well executed and reach convincing, if limited, conclusions.

However, the authors face serious technical hurdles which limit their ability to obtain meaningful, compelling evidence that would help to establish the nature of RV:DC host relationships. The main issue is the (unexplained and uninvestigated) extremely low infection rate of mDCs, which could indicate that these cells are not targeted by RV in infected hosts. The authors must provide an explanation of their findings of relative resistance.

Since the paper addresses an important and interesting topic, the authors are encouraged to revisit their studies, extend their approach, and reach a more compelling conclusion regarding the relationship of RV with host DCs.

Specific comments/concerns:

1. The extremely low infection rate in mDCs is a concern, hampering mechanistic analyses. The findings could indicate that DCs are not targeted by RV at all upon initial replication at the portal of entry, calling into the question the entire premise for the study. Viruses known to target DCs at the portal of entry (HIV, polio, dengue, VEE, etc.) readily will infect mDCs with high efficiency in vitro. At a minimum, the authors should discuss this possibility. In this reviewer’s opinion, the low infection rate will make it extremely difficult to reliably assess the RV:DC relationship in the mDC model.

2. The study lacks an investigation of the innate antiviral response. If RV targets DCs for infection, their ability to mount innate antiviral defenses should be investigated rigorously, eg. with detailed analyses of the innate signaling response.

3. The authors use as simplistic concept of “DC activation” and its role in adaptive antiviral immunity. DC “activation” is not a switch that is turned on upon pathogen exposure, but rather a series of carefully orchestrated signaling responses that coordinate induction of maturation markers (eg. CD80/86), LN homing receptors (eg. CCR7), the immuno-proteasome, peptide antigen trans-locators (TAP1/2), the peptide loading complex (tapasin:ERp57:MHC I), etc. These adaptive responses can lead to cross-presentation of viral antigen and priming of antiviral CD8 T cell immunity. Cross presentation/T cell priming are major targets for viral immune suppression strategies: even in circumstances where cytokines/maturation markers are induced (such as in the present case), other events (esp. antigen presentation/MHC I, DC viability) may be inhibited.

The authors’ data on maturation markers/CCR7 induction in RV-infected mDCs indicate a certain activation state, but by no means suggests that these DCs are engaged in adaptive antiviral immunity (many other events important in cross presentation/T cell priming may be suppressed). This may explain that signs of DC engagement in vitro clashed with lacking DC function, eg. in LN homing/migration.

4. The comparisons with LPS are not helpful. LPS (TLR4 agonist), in the manner used here, is not a realistic or informative positive control for DC activation phenotypes. It is a “sledgehammer” inducer of innate signaling, with extreme, artificial, non-physiologic effects on myeloid cells. Since LPS exerts lethal toxicity in vivo, it is not a useful side-by-side comparator to more realistic, natural challenges of myeloid cells, such as (possible) RV infection.

Reviewer #3: Conclusions are not supported by the data in its present form (see summary)

**Editorial and Data Presentation Modifications?**

Reviewer #1: (No Response)

Reviewer #2: See main review

Reviewer #3: (No Response)

**Summary and General Comments**

Reviewer #1: In the present study, the authors investigated the in vitro response of human monocyte-derived dendritic cells (moDCs) to different strains of rabies virus. The findings indicate that RABV strains exhibit variability in their capacity to infect and activate moDCs in vitro. Nevertheless, exposure to any RABV strain did not result in a notable antiviral status or the inhibition of moDC reactivity. This study offers insights into the manner in which rabies virus interacts with cells (DCs) that are pivotal for early immune responses, and underscores the necessity for further research to enhance immune activation by targeting dendritic cells. I believe that this study has implications for the field of rabies research; however, there are still some questions that require further investigation.

1. The three distinct RABV strains illustrated in Figure 1C were inoculated into moDC and SH-SY5Y cells. All three strains demonstrated successful infection with moDC at a multiplicity of infection (MOI) of 5, which yielded the highest viral titer. However, it is unclear why MOI 5 was not employed for infection with SH-SY5Y cells.

2. The inability of moDCs exposed to RABV to elicit T cell proliferation in vitro is depicted in Figure 3. How was the underlying cause of the observed deficiency in T cell proliferation ascertained in this experiment? Was this caused by RABV infection, or caused by the RABV-treated DC cells?

3. This study primarily assessed the impact of various RABV strains on moDC. Should the authors evaluate the activation levels of MHC-I and MHC-II in response to different RABV strains following moDC treatment, in order to enhance the overall data quality?

Reviewer #2: See main review

Reviewer #3: This is a very interesting study by Kroh and colleagues, which explores the response to monocyte-derived DCs to Rabies virus infection using three different Rabies strains. The study has a lot of potential and very interesting results already, such as the fact that one of the strains does not infect DCs at all. However, there are some major issues in the opinion of this reviewer that require further insight. In its present form, I feel that the conclusions of the study are not supported by the data presented. Specific comments are as follows:

1- MoDCs represent only one subset of inflamatory DCs derived from monocytes (not from the pre-DC lineage) so, data cannot be extrapolated to DCs in general. The response of other DC subsets to Rabies virus infection may be completely different. For the same reasons, caution should be used by the authors when they extrapolate their findings to explain the T cell response to Rabies virus

2- The T-cell data is confusing for this reviewer. I understand from the methods section that the T cells utilized for the readout are autologous but not Rabies specific or even antigen-experienced T cell pools. I understand that these T cells proliferate in response to Dynabeads but I cannot expect these cells to proliferate in response to presentation of Rabies antigen if they are naïve, even if the moDCs are doing their job really well.

3- Related to this, the authors are exposing T cells to a mass population of moDCs in which, in the best case, only 10-15% of cells are infected. I know this is note easy, but what would happen if they used rabies infected sorted DCs?

4- For the reasons explained in point 2, I don´t think that the conclusion of Rabies virus activating moDCs but not leading to virus-specific T cell responses is correct. I think that, in order to conclude this, the authors would need to demonstrate that the Rabies exposed moDCs are not able to induce proliferation of Rabies-experienced T cells.

Minor

1- In the CFSE proliferation experiments it would be good to see the peaks of T cell proliferation (as a readout of CFSE dilution) rather than a bulk of cells that have lost CFSE fluorescence. That would help to compare and assess levels of T cell proliferation quantitatively.

2- Is MFI median or mean fluorescence intensity? Median fluorescence intensity allows to better understand comparatively the levels of expression of the majority of cells in a mass population

3- LPS by itself stimulates T cell proliferation (see Tough et al., JEM 1997), so in Figure 3 any LPS carried out in the co-culture by the previous DC stimulation can do the job without DCs.

4- Can any virus expressing DC antagonist proteins suppress the stimulating effect of LPS?

PLOS authors have the option to publish the peer review history of their article (what does this mean? ). If published, this will include your full peer review and any attached files.

**Do you want your identity to be public for this peer review?** For information about this choice, including consent withdrawal, please see our Privacy Policy .

Reviewer #1: No

Reviewer #2: No

Reviewer #3: No

---

## [Decision Letter · Decision Letter 1]

17 Mar 2025

Dear Dr. Embregts,

We are pleased to inform you that your manuscript 'A comparative analysis of the dendritic cell response upon exposure to different rabies virus strains' has been provisionally accepted for publication in PLOS Neglected Tropical Diseases.

Best regards,

Elvina Viennet, PhD

Section Editor

Elvina Viennet

Section Editor

Shaden Kamhawi

co-Editor-in-Chief

Paul Brindley

co-Editor-in-Chief

Reviewer's Responses to Questions

**Key Review Criteria Required for Acceptance?**

**Methods**

-Are the objectives of the study clearly articulated with a clear testable hypothesis stated?

-Is the study design appropriate to address the stated objectives?

-Is the population clearly described and appropriate for the hypothesis being tested?

-Is the sample size sufficient to ensure adequate power to address the hypothesis being tested?

-Were correct statistical analysis used to support conclusions?

-Are there concerns about ethical or regulatory requirements being met?

Reviewer #3: (No Response)

Reviewer #4: (No Response)

Reviewer #5: The study's objective is to evaluate the condition and characteristics of moDC when exposed to different RABV infections.The authors measured the replication process of RABV in moDC, followed by a T cell activation assay.It was observed that there were some differences in functionality between dogRV, SHBRV and SAD5 RABV vaccine strain infected moDC.However, neither SHBRV nor SAD5 infected moDC were able to activate T cells after exposure, while maintaining its functionality. The study's hypothesis is clearly articulated and the methods and approaches are adequate to address the testable hypothesis.

Reviewer #6: The study clearly states its objectives and hypothesis, focusing on the interaction of different rabies virus (RABV) strains with human monocyte-derived dendritic cells (moDCs). The rationale is well justified given the known immune evasion mechanisms of RABV. The use of in vitro models to study moDC responses to RABV exposure is appropriate and aligns with the stated objectives. The comparison of three RABV strains strengthens the analysis. The manuscript details the use of six individual donors for moDCs, ensuring biological variability. While this sample size is reasonable, additional justification for the selected number of donors regarding statistical power would be beneficial and appropriate statistical methods were used, including Friedman tests for non-parametric comparisons. However, clarification on adjustments for multiple comparisons would improve transparency.

**Results**

-Does the analysis presented match the analysis plan?

-Are the results clearly and completely presented?

-Are the figures (Tables, Images) of sufficient quality for clarity?

Reviewer #3: (No Response)

Reviewer #4: (No Response)

Reviewer #5: The results are well regarded, and I would like to humbly suggest a few potential amendments that could further illuminate the findings.

1. Fig. 1C: It would be greatly appreciated if the authors could provide a statistical assessment of RABV titer at different hours post infection (e.g. 0, 24, 48). It might be beneficial to ascertain whether a statistically significant increase in virus replication is observed, as this could provide crucial evidence for concluding that replication is indeed occurring. Additionally, could the authors kindly provide some explanation on the timeline of the experiment? In particular, could they explain why only 24 and 48 hours post-infection time points were tested? Given the low infection rate and slow RABV kinetics, was there enough time for moDC to change the phenotype and start presenting antigen?

Reviewer #6: The results are presented in alignment with the study objectives and the analytical plan. The figures and flow cytometry analyses adequately support the findings. Data are clearly presented with appropriate use of figures and tables. However, some figures (e.g., cytokine concentration heatmaps) could be more refined for readability.

Figures are well-structured, but some images could be improved in resolution, especially for immunofluorescence microscopy.

**Conclusions**

-Are the conclusions supported by the data presented?

-Are the limitations of analysis clearly described?

-Do the authors discuss how these data can be helpful to advance our understanding of the topic under study?

-Is public health relevance addressed?

Reviewer #3: (No Response)

Reviewer #4: (No Response)

Reviewer #5: The conclusions appear to be supported by the data. I would like to suggest a few minor comments for your consideration:

1. It would be greatly appreciated if you could discuss the correlation between your previous findings using transcriptomics of RABV infected moDC with more details. If moDCs are activated by RABV infection, could you please explain/hypothesize what the roadblocks are that prevent antigen presentation and T cell activation?

2. It is understood that LPS is a very strong activation signal, and the level of response upon LPS exposure is extremely high. In this context, measuring the functionality of the moDC after RABV infection using LPS may still give positive results even if the moDC state were affected by RABV infection.

3. The authors may like to elaborate on the genotype/phenotype difference between DogRV, SHRBV and SAD5 to better understand the difference in moDC response.

Reviewer #6: The conclusions logically follow the presented data, emphasizing the failure of moDCs to fully activate upon RABV exposure. The authors acknowledge limitations, including the use of an in vitro model that may not fully capture in vivo immune interactions. Further exploration of transcriptomic data could add depth to the analysis.

**Editorial and Data Presentation Modifications?**

Reviewer #3: (No Response)

Reviewer #4: (No Response)

Reviewer #5: I would like to recommend a minor revision with minor modification of existing data.

Reviewer #6: The manuscript is well-written, though minor grammatical refinements could improve readability.

Some figures could be optimized for clarity (e.g., immunofluorescence images should be presented with improved contrast for better visualization of RABV-N staining).

The citation list is comprehensive, but recent studies on dendritic cell immune evasion mechanisms could be incorporated.

**Summary and General Comments**

Reviewer #3: (No Response)

Reviewer #4: The study by Kron and colleagues examining the effects of different rabies virus strains on monocyte-derived dendritic cells is compelling and well-executed, with meticulously presented data and thoughtful discussion. Within the defined scope of investigating monocyte-derived DCs as a primary model system, the research provides valuable insights into viral-host cell interactions.

The results effectively characterize the DC response patterns within their experimental parameters. The authors discuss the potential roles of conventional DC1s and DC2s in their discussion section, and it would be particularly interesting to also consider Langerhans cells, given their unique position as sentinel cells in the skin - the typical entry point for rabies virus infection. The authors provide clear and well-reasoned justification for focusing specifically on monocyte-derived DCs, acknowledging both the advantages and limitations of this approach.

The methodological choices are explained, strengthening the validity of their conclusions in this investigation of DC-rabies virus interactions.

Reviewer #5: The study address an interesting question about the RABV biology and aims to understand the role of moDC during infection. More importantly, this study reveals that the functionality of our adopted immune system may be impaired by the virus. Overall, the study provides an additional data indicating the role of moDC in RABV infection and its significance. The study will benefit from animal data showing the lack of T cell activation upon dogRV versus RABV SAD5 infection.

Reviewer #6: This study provides valuable insights into how different RABV strains interact with moDCs, shedding light on the failure to mount an effective adaptive immune response. Strengths include the comparative approach using three RABV strains and the detailed immunophenotypic analysis. However, limitations include the relatively small sample size and the lack of in vivo validation.

Recommendations:

Justify the sample size with a statistical power analysis.

Improve the resolution of immunofluorescence images.

Provide more details on adjustments for multiple comparisons in statistical analyses.

Discuss potential implications of these findings for vaccine development and immunotherapeutic strategies.

PLOS authors have the option to publish the peer review history of their article (what does this mean? ). If published, this will include your full peer review and any attached files.

**Do you want your identity to be public for this peer review?** For information about this choice, including consent withdrawal, please see our Privacy Policy .

Reviewer #3: No

Reviewer #4: No

Reviewer #5: **Yes: ** Alexander Malogolovkin

Reviewer #6: **Yes: ** Iana S. S. Katz

---

## [Editor Report · Acceptance letter]

Dear Dr. Embregts,

We are delighted to inform you that your manuscript, "A comparative analysis of the dendritic cell response upon exposure to different rabies virus strains," has been formally accepted for publication in PLOS Neglected Tropical Diseases.

Best regards,

Shaden Kamhawi

co-Editor-in-Chief

Paul Brindley

co-Editor-in-Chief
